# Was the Reduction in Seasonal Influenza Transmission during 2020 Attributable to Non-Pharmaceutical Interventions to Contain Coronavirus Disease 2019 (COVID-19) in Japan?

**DOI:** 10.3390/v14071417

**Published:** 2022-06-28

**Authors:** Keita Wagatsuma, Iain S. Koolhof, Reiko Saito

**Affiliations:** 1Division of International Health (Public Health), Graduate School of Medical and Dental Sciences, Niigata University, Niigata 951-8510, Japan; jasmine@med.niigata-u.ac.jp; 2Japan Society for the Promotion of Science, Tokyo 102-0083, Japan; 3College of Health and Medicine, School of Medicine, University of Tasmania, Hobart 7000, Australia; iain.koolhof@health.tas.gov.au

**Keywords:** COVID-19, SARS-CoV-2, seasonal influenza, NPIs, epidemics

## Abstract

We quantified the effects of adherence to various non-pharmaceutical interventions (NPIs) on the seasonal influenza epidemic dynamics in Japan during 2020. The total monthly number of seasonal influenza cases per sentinel site (seasonal influenza activity) reported to the National Epidemiological Surveillance of Infectious Diseases and alternative NPI indicators (retail sales of hand hygiene products and number of airline passenger arrivals) from 2014–2020 were collected. The average number of monthly seasonal influenza cases in 2020 had decreased by approximately 66.0% (*p* < 0.001) compared to those in the preceding six years. An increase in retail sales of hand hygiene products of ¥1 billion over a 3-month period led to a 15.5% (95% confidence interval [CI]: 10.9–20.0%; *p* < 0.001) reduction in seasonal influenza activity. An increase in the average of one million domestic and international airline passenger arrivals had a significant association with seasonal influenza activity by 11.6% at lag 0–2 months (95% CI: 6.70–16.5%; *p* < 0.001) and 30.9% at lag 0–2 months (95% CI: 20.9–40.9%; *p* < 0.001). NPI adherence was associated with decreased seasonal influenza activity during the COVID-19 pandemic in Japan, which has crucial implications for planning public health interventions to minimize the health consequences of adverse seasonal influenza epidemics.

## 1. Introduction

Seasonal influenza viruses are major causes of respiratory infections worldwide. Severe acute lower respiratory tract infections contribute to hospitalizations and mortality in both children and adults and represent a major threat to public health owing to annual epidemics and the potential to cause pandemics [1]. Globally, in 2018, among children under 5 years of age, there were an estimated 109.5 million influenza virus infections, 0.9 million hospital admissions, and up to 34,800 overall influenza-associated deaths [2]. Moreover, epidemics affect economic activity and, in some instances, social stability [3]. Generally, the epidemiology of influenza is seasonally driven, with the majority of transmission occurring during winter in temperate zones, including Japan [4]. Monitoring circulation patterns of seasonal influenza viruses is an essential component of the annual planning for national prevention and response activities in countries worldwide.

The novel severe acute respiratory syndrome coronavirus 2 (SARS-CoV-2), the causative virus of coronavirus disease (COVID-19) was first reported in Wuhan, China, at the end of 2019. Following which, the COVID-19 pandemic rapidly became an unprecedented global health threat, which continues today [5]. By the end of 2020, more than 82.6 million confirmed cases were reported worldwide, causing more than 1.8 million deaths, resulting in a substantial burden of disease [6]. To reduce the transmission of respiratory infections, including COVID-19, it is crucial to adopt non-pharmaceutical interventions (NPIs), encompassing personal protective measures (e.g., hand hygiene and face masks), social distancing measures (e.g., isolation of the ill and quarantine of exposed individuals, prevention of mass gatherings, and school and workplace closures), and travel-related measures (e.g., travel restrictions and border closures), especially before an effective specific intervention (e.g., vaccine conferring long-lasting protective immunity) becomes widely available [7,8,9,10]. The World Health Organization has recommended NPIs to reduce the transmissibility of COVID-19 [11]. Indeed, the potential effect of these NPI strategies provides a valuable opportunity to promote and assess universal disease control and prevention measures in community settings.

Most recently, the transmission of seasonal influenza has been described to have dramatically declined in European countries, the United States (US), China, and Japan during the COVID-19 pandemic, with seasonal influenza infections in 2020 being lower than those in preceding epidemiologic years [12,13,14,15,16]. This is further reflected in countries in the Southern Hemisphere, such as Australia and South Africa, where no historic summer epidemics have been observed and the incidence of seasonal influenza infections has remained relatively low [17,18]. This reduction in influenza has hindered the ability to provide information on circulating viruses to inform annual seasonal influenza vaccine strain selection. Remarkably, the adoption of NPIs aimed at reducing the transmissibility of COVID-19 may also contribute to variations in the epidemic dynamics of other respiratory infections, including seasonal influenza viruses and human respiratory syncytial viruses (HRSVs) [19,20]. Nevertheless, quantitative evidence of changes in the transmission dynamics of seasonal influenza in relation to these public health interventions during the COVID-19 pandemic in Japan is limited and unclear.

Th present study aimed to analyze sentinel surveillance data of seasonal influenza and alternative indicators of NPIs in Japan to critically assess the potential effect of changes in adherence to various NPIs on the transmission dynamics of seasonal influenza after the COVID-19 pandemic started in 2020. By exploring the associations between indicators of NPIs and seasonal influenza transmission, it is possible to evaluate the underlying mechanisms of transmission and provide insights for current and future public health management. Indeed, a detailed understanding of the epidemiological dynamics of seasonal influenza viruses in this long series could be fruitful for understanding temporal changes in the transmissibility of infectious diseases. Here, we first compared the incidence of seasonal influenza infections in Japan in 2020 to that in the preceding six epidemiological years (2014–2019). We then quantified the time-series monthly associations between the incidence of seasonal influenza infections and adherence to alternative indicators of NPIs, including the retail sales of hand hygiene products and the number of international and domestic airline passenger arrivals in Japan over the study period (2014–2020).

## 2. Materials and Methods

### 2.1. National Seasonal Influenza Surveillance Data

The notifications of influenza used in this study were obtained from the Infectious Disease Weekly Report (IDWR), which was sourced from the National Epidemiological Surveillance of Infectious Diseases (NESID) data published by the National Institute of Infectious Diseases, Japan (NIID) under the Ministry of Health, Labor and Welfare, Japan (MHLW) [21]. The MHLW manages approximately 5000 sentinel sites (i.e., hospitals and clinics) in Japan, which reports the number of patients diagnosed with a seasonal influenza infection on a weekly basis to the prefectural or municipal public health sectors in Japan. Seasonal influenza falls under the sentinel surveillance arm of the program [22]. Generally, the number of sentinels designated to each public health service area is determined based on population size: a public health center with <30,000 individuals has one sentinel; a center with 30,000–75,000 individuals has two sentinels; and one with >75,000 individuals has ≥3 sentinels, as determined by the following formula: 3 + (population-75,000)/50,000 [20,23]. These sentinels use the following criteria to notify patients of seasonal influenza-like illness (ILI): (1) sudden onset of illness, (2) fever >38 °C, (3) symptoms of upper respiratory inflammation, and (4) systemic symptoms, such as general fatigue. A confirmed case of seasonal influenza infection was defined by the reporting criteria if the patient met all symptoms from (1) to (4) or had at least one of the symptoms combined with a positive rapid diagnostic test kit [22]. These sentinel sites forward clinical data to approximately 60 prefectural or municipal public health sectors, which were then electronically transferred to the NIID, where the number of seasonal influenza cases is released weekly through its website (https://www.niid.go.jp/niid/en/idwr-e.html, accessed on 25 May 2022). To elucidate the epidemiological dynamics of seasonal influenza in Japan, we extracted the total number of seasonal influenza cases per sentinel site at the national level in Japan (hereafter referred to as “seasonal influenza activity”) recorded in weeks 1–52 in 2014–2020 from the NESID, and compiled the monthly seasonal influenza activity based on these weekly data. The monthly seasonal influenza activity for the seven epidemiologic years studied (2014–2020) was used as the dependent variable in the time-series statistical models presented here.

### 2.2. Alternative Indicators of NPIs

#### 2.2.1. Retail Sales of Hand Hygiene Products

To assess the level of personal hand hygiene behavior (i.e., potential effect of sanitary measures), we used data on the monthly retail sales of hand hygiene products (i.e., hand soap and ethyl alcohol) per ¥1 billion (approximately £6,800,000/$9,000,000) (units: yen) at the national level in Japan for seven epidemiologic years (2014–2020) as an independent variable in the time-series statistical models presented here. These data were extracted from the statistics of production in chemical industries under the Ministry of Economy, Trade, and Industry, Japan (METI) [24]. In view of the missing published data on the monthly number of hand hygiene products for long time-series at national level in Japan and the comparability with several previous literatures conducted in Japan, we utilized retail sales of hand hygiene products as a proxy variable associated with seasonal influenza activity [20,25]. Unfortunately, we could not also obtain a dataset on the retail sale of masks by month; therefore, we did not incorporate this indicator into the analyses.

#### 2.2.2. International and Domestic Airline Passenger Arrivals Data

To assess the level of travel restrictions, we used the monthly number of international and domestic airline passenger arrivals per one million population (units: person) at the national level in Japan for the seven epidemiologic years (2014–2020) as an independent variable in the time-series statistical models presented here. These data were extracted from the statistics of air transport from the Ministry of Land, Infrastructure, Transport and Tourism, Japan (MLIT) [26].

#### 2.2.3. Meteorological Data

Meteorological driving factors, such as the average ambient temperature (units: °C) and relative humidity (units: %), are thought to be associated with the dynamics of seasonal influenza transmission in temperate regions [27,28,29]. Therefore, we used monthly meteorological data (i.e., average ambient temperature and relative humidity) published by the Japan Meteorological Agency as independent variables in the time-series statistical models presented herein [30]. Monthly meteorological data collected from meteorological observatories located in the prefectural capital city were used for each prefecture. We extracted the monthly average ambient temperature and relative humidity for seven epidemiologic years (2014–2020). Two meteorological observatories had missing data on relative humidity (Saitama and Shiga Prefectures); therefore, we selected observatories at the nearest distance (≤50 km) from the prefectural capital by substitution. The average temperature and relative humidity over Japan were calculated using monthly meteorological data for each prefecture.

### 2.3. Statistical Analysis

#### 2.3.1. Descriptive Statistics

Descriptive statistics were first assessed to clearly describe the temporal dynamics of the multitudes of epidemiological datasets. We visually assessed the trends of time-series variations in seasonal influenza activity, alternative indicators of NPIs (i.e., retail sales of hand hygiene products and number of international and domestic airline passenger arrivals), and meteorological conditions (i.e., average ambient temperature and relative humidity) during the study period (2014–2020) to assess the basic characteristics of the included dataset. Additionally, the mean and standard deviation (SD) of these dependent and independent variables are described. To quantify the transmission dynamics of seasonal influenza in 2020 in Japan, we compared the monthly seasonal influenza activity in 2020 with the average seasonal influenza activity in the corresponding period in the six preceding epidemiologic years (2014–2019) using a monthly paired *t*-test.

#### 2.3.2. Identification Strategy of the Time-Series Statistical Regression Model

We next formulated a time-series association between monthly seasonal influenza activity and long-term exposure to alternative indicators of NPIs at the national level in Japan during the study period (2014–2020) by running a time-series statistical model for seasonal data. Several steps were taken to build a robust and reliable statistical model. First, we adopted an independent approach using the alternative indicators of NPIs and meteorological conditions as independent variables when relating the alternative indicators of NPIs to seasonal influenza activity. Prior to constructing the model, we checked the probability distribution of the dependent variable, the monthly seasonal influenza activity (the normality of probability distribution was assessed by the Shapiro–Wilk test) (Appendix A), and assessed the linearity between seasonal influenza activity and each independent variable. All independent variables included in the statistical model were assessed for multicollinearity using the pairwise Spearman’s rank-order cross-correlation coefficient (*ρ*). If the variables were found to be highly linearly correlated (cut-off of |*ρ*| > 0.8), the variable with the largest mean absolute statistical correlation with the other independent variables was removed [31]. Based on the preliminary analysis results, no independent variables showing strong statistical linear correlations were observed (Appendix A).

To address the delayed heterogeneity of meteorological variables on the dynamics of seasonal influenza transmission, we considered temporal lags (i.e., delays in potential effect) of up to 4 months from several previous studies [20,27,32,33,34,35,36]. We used standard time-series generalized linear regression models (GLMs) with a gamma distribution and natural logarithmic link function. Robust error variances between the monthly seasonal influenza activity and single variables of meteorological conditions (average ambient temperature and relative humidity) with lags of 0–4 months were assessed based on the Akaike information criterion (AIC) [37]. Generally, the AIC is described as –2 log (*L*) + 2 *K* where log (*L*) is the maximum value of the natural logarithmic-likelihood function of the statistical model and *K* represents the number of parameters. Based on the preliminary analysis results, we considered meteorological conditions with optimal delayed effects (i.e., the average ambient temperature at lag 0–1 months and the relative humidity at lag 0 months), which minimized the AIC, and developed a core statistical model (Appendix A). After establishing the optimal lag setting of meteorological variables in the core model, different lag structures in the associations between monthly seasonal influenza activity and NPI indicators were explored using single lag months (lag 0, 1, 2, 3, and 4 months) and cumulative lag months (lag 0–1, 0–2, 0–3, and 0–4 months moving average). The cumulative lag months were considered to extract the overall trend of the time series without being influenced by the month time-dependent noise. Indeed, assessments of temporal lags before modeling were used to account for the biological processes and natural history of the virus, host reservoir population dynamics, and the infection incubation periods prior to the onset of symptoms of notified cases.

We then formulated a standard time-series GLM with a gamma distribution and logarithmic link function with robust error variances, allowing for overdispersion, to critically investigate the time-series association between monthly seasonal influenza activity and alternative indicators of NPIs at the national level in Japan; this was conducted while adjusting for meteorological conditions with optimal delayed effects (i.e., average ambient temperature at lag 0–1 months and relative humidity at lag 0 months) and seasonal variations and cycles of infectious diseases. Specifically, monthly seasonal influenza activity (continuous variable) was the dependent variable, and monthly retail sales of hand hygiene products per ¥1 billion (continuous variable) at the national level in Japan during 2014–2020, monthly number of international and domestic airline passenger arrivals per one billion populations (continuous variable) at the national level in Japan during 2014–2020, and meteorological conditions with optimal delayed effects (continuous variable) were included as independent variables driving the dynamics of seasonal influenza transmission. The statistical model was adjusted using year variables (2014, 2015, 2016, 2017, 2018, 2019, and 2020) (categorical variables) as covariates to control for long-term seasonal variations and cycles of infectious diseases. One year was defined as January–December. Additionally, as autocorrelation of residuals in the case of infectious disease is pathogen-specific and needs to be accounted for, autoregressive terms of order one (i.e., one month) were incorporated into the statistical models [38]. The goodness-of-fit of the statistical model was assessed in a combined manner using the dispersion parameter (*α*) and AIC [37,39]. Generally, *α* is the variance parameter of the model, with an *α* value of <1.5 suggesting that the deviation of the observed data from the model is not too large (i.e., the model fits the observed data well). Although there is no theoretical basis for this criterion, it has been reported that *α* < 1.5 can significantly improve the degree of overdispersion [39,40]. In the present analysis, the overdispersion of each model was mitigated using gamma distribution to ensure that *α* was <1.5. Generally, the best model with a lower AIC value is preferred, as it achieves a better combination of goodness-of-fit and parsimony.

Formally, the general algebraic definition of time-series statistical models is as follows:(1)$$P\left(y_{i,t}|e_{i,t}\right)\propto Gamma\;\left(e_{i,t}\right),\;\log\;\left(e_{i,t}\right)≃\alpha +\beta x_{i,t}+\underset{j}{\sum }f_{G}\left(z_{i,j,t}\right)+\phi_{i,t}+e_{i,t-1}+\sin\left(2\pi \cdot \frac{\theta_{i,t}}{12}\right)+\cos\left(2\pi \cdot \frac{\theta_{i,t}}{12}\right)+\epsilon_{i,t}$$
where *y_i_*_,*t*_ denotes the outcome time-series; *e_i_*_,*t*_ denotes the expected time-series of the monthly seasonal influenza activity in month *t* (*t* = 1, 2, *…*); *x_i_*_,*t*_ denotes the *i*th alternative indicator of NPIs (the retail sales of hand hygiene products and the number of international and domestic passenger arrivals) with different lag strictures, including each single lag month and cumulative lag months for monthly seasonal influenza activity (*i* = 1, 2, *…*); *f_G_* (*z_i_*_,*j*,*t*_) denotes the *jth* meteorological variables (the average temperature and the relative humidity) with optimal delayed effects based on minimized AIC, respectively (*j* = 1, 2, *…*); *ϕ_i_*_,*t*_ denotes indicator variables of year; *e_i_*_,*t−*1_ denotes autoregressive terms at order one, accounting for potential serial correlation; and the statistical model includes the periodic harmonic functions called the Fourier series, which is formed by the sum of sines and cosines of month *θ_i_*_,*t*_ [41,42]. The Fourier series terms can be used to recreate any periodic signal (such as a consistent seasonal pattern) using a linear combination of sine and cosine waves of varying wavelengths. Indeed, the *ϕ_i_*_,*t*_, *e_i_*_,*t−*1_, and Fourier series terms were included to capture the long- and short-series seasonal variations and cycles of infectious diseases and to minimize residual autocorrelations of infectious diseases, respectively. The linear term α corresponds to the overall intercept, the linear term *β* indicates adjusted linear regression coefficients (continuous or categorical variables with adjusted linear regression coefficients), and the term *ε_i_*_,*t*_ corresponds to the intercept. Indeed, by including all variables of interest in the same regression equation, we strengthen the interpretation of the effects as independent and additive, based on accumulated empirical knowledge.

In the present study, a series of time-series statistical models was developed to assess the direct effects of each alternative indicator of NPIs on monthly seasonal influenza activity. The maximum likelihood method was used for the model. To examine the robustness of the main findings, we performed sensitivity analyses using an extended study period until December 2021 to account for the long-term effects of alternative indicators of NPIs on monthly seasonal influenza activity (i.e., the latest date for which the all variables of dataset are available in Japan as of June 2022). Statistical significance was set at a *p*-value of <0.05 (type I error), on a two-tailed test. All statistical analyses were performed using STATA version 15.1 statistical software (Stata Corp, College Station, TX, USA).

### 2.4. Ethical Considerations

The present time-series ecological modeling study analyzed publicly available data. The datasets used in our study were de-identified and fully anonymized in advance, and, as such, the analysis of publicly available data without any identifiable information did not require ethical approval. The present study was conducted in accordance with the Declaration of Helsinki (revised in 2013).

## 3. Results

### 3.1. Descriptive Description

We first performed a descriptive analysis of the monthly time-series variations in the total number of seasonal influenza cases per sentinel site at the national level in Japan (i.e., seasonal influenza activity), alternative indicators of NPIs (retail sales of hand hygiene products and number of international and domestic airline passenger arrivals), and meteorological conditions (average ambient temperature and relative humidity) at the national level in Japan during 2014–2020 (Figure 1 and Appendix A). To better understand the temporal dynamics, the seasonal influenza activity for seven epidemiologic years (2014–2020) is displayed in Figure 1A, illustrating an abrupt decrease during the 2020 COVID-19 pandemic in Japan. Before the COVID-19 pandemic, the mean monthly seasonal influenza activity was relatively similar between years: 29.30 cases in 2014, 19.75 cases in 2015, 29.42 cases in 2016, 27.08 cases in 2017, 31.92 cases in 2018, and 31.50 cases in 2019. However, 2020 showed the lowest activity, with 9.51 cases (Appendix A). Focusing on the epidemic curve for 2014 and 2017, it was observed that peaks of seasonal influenza activity were relatively low compared to other epidemiologic years (i.e., 2015, 2016, 2018, and 2019). However, according to NIID, the peaks for weekly number of out-patient visits in 2014 (34.4 visits/clinic/week) and 2017 (39.4 visits/clinic/week) were not different from the mean for 7 epidemiologic years (mean: 39.7 visits/clinic/week; SD: 13.1 visits/clinic/week), suggesting the epidemic in the two years were within seasonal range [43]. The average monthly seasonal influenza activity in 2020 in Japan was estimated to have decreased by approximately 66.0% (monthly paired *t*-test, *p* < 0.001) compared to those in the preceding 6 epidemiologic years (2014–2019).

Interestingly, during the study period (2014–2020), the variations in the alternative indicators of NPIs (retail sales of hand hygiene products and number of international and domestic airline passenger arrivals) showed distinct changes around 2020 (Figure 1B–D). The monthly retail sales of hand hygiene products remained broadly consistent at approximately ¥7 billion (£47,600,000/$63,000,000) for the preceding 6 epidemiologic years (2014–2019), but showed a sharp upward trend in 2020, steadily remaining at approximately ¥10 billion (£68,000,000/$90,000,000) (Figure 1B and Appendix A). During the preceding 6 epidemiologic years (2014–2019), the monthly number of domestic and international airline passenger arrivals was approximately 8.0 million and 1.7 million, respectively, before falling sharply to approximately 3.8 million and 0.3 million, respectively, in 2020 (Figure 1C,D and Appendix A). In contrast, there was a clear annual seasonality and cycle in the series of variations in meteorological conditions, with the mean average ambient temperature and relative humidity throughout Japan being approximately 17 °C and 71%, respectively; however, there was no marked change in 2020 (Figure 1E,F and Appendix A). The detailed numerical statistics by year are presented in Appendix A.

### 3.2. Identifying the Association between Seasonal Influenza Transmission and Alternative Indicators of NPIs

Figure 2 displays the overall percentage changes in monthly seasonal influenza activity associated with per 1-unit increase in each alternative indicator of NPIs (the retail sales of hand hygiene products and the number of international and domestic airline passenger arrivals) with different lag structures selected a priori (i.e., the single lag months (lag 0, 1, 2, 3, and 4 months) and the cumulative lag months (lag 0–1, 0–2, 0–3, and 0–4 months moving average)) based on the standard time-series GLMs with gamma distribution and logarithmic link function with robust error variances. After controlling for the effect of potential confounders, we observed a negative overall association between monthly seasonal influenza activity and retail sales of hand hygiene products, and a positive association between international and domestic airline passenger arrivals with monthly seasonal influenza activity and throughout different lag structures in each statistical model (Figure 2 and Appendix A). Specifically, we found that for every ¥1 billion spent on retail sales of hand hygiene products, there was a 7.0% to 4.5% decrease in seasonal influenza notifications in between 0- to 4-month lags (Figure 2A and Appendix A). An increase of one million domestic and international airline passenger arrivals was found to increase cases of seasonal influenza by 5.3–8.8% and 20.5–29.3%, respectively, in lags from 0 to 4 months (Figure 2B,C and Appendix A).

Subsequently, we assessed the shape of the cumulative lag effect as an alternative indicator of NPIs in relation to monthly seasonal influenza activity (Figure 2 and Appendix A). We found that the effects generally increased within a 3-month lag in retail sales of hand hygiene products associated with seasonal influenza activity (varied from approximately 15.5% to 7.8%) (Figure 2A and Appendix A). The largest cumulative lag effect estimate for all groups within retail sales of hand hygiene products was at lag 0–3 months, with a significant negative association of –15.5% (95% confidence interval [CI]: 20.0–10.9%; *p* < 0.001; AIC = 4.01; *α* = 0.95) in monthly seasonal influenza activity for every ¥1 billion increase in retail sales of hand hygiene products. This suggests that the medium- to long-term effects of sanitary measures are relatively large (approximately 3 months). However, the monthly seasonal influenza activity attributable to the cumulative lag structure of airline passenger arrivals remained constant and with minimal variation (varied from approximately 9.5% to 11.6% for domestic airline passenger arrivals from 27.8% and 30.9% for international airline passenger arrivals, respectively). Specifically, an increase in the average of one million domestic and international airline passenger arrivals had a significant positive association with seasonal influenza activity by 11.6% at lag 0–2 months (95% CI: 6.70–16.5%; *p* < 0.001; AIC = 3.99; *α* = 1.12) and 30.9% at lag 0–2 months (95% CI: 20.9–40.9%; *p* < 0.001; AIC = 4.12; *α* = 0.89) with the largest cumulative lag effect (Figure 2B,C and Appendix A). By assessing these differently shaped single and cumulative lag structures, we were able to quantify the range of immediate and delayed effects of the alternative indicators of NPIs associated with seasonal influenza activity and explore detailed exposure–response relationships.

### 3.3. Further Investigations

In addition, we conducted sensitivity analyses to verify the robustness of our main findings. It should be noted that the time-series association between seasonal influenza activity and alternative indicators of NPIs at the national level in Japan did not change substantially and remained robust when the study period was expanded until December 2021 (Appendix A).

## 4. Discussion

The present study aimed to quantitatively assess the potential causal effect of changes in adherence to various NPIs (retail sales of hand hygiene products and number of international and domestic airline passenger arrivals) for COVID-19 suppression on the incidence of seasonal influenza infections at the national level in Japan. Despite these simplified assumptions, our findings objectively suggest that the adoption of NPIs at the national level in Japan may have had a beneficial effect on reducing the transmission of seasonal influenza viruses. Indeed, the total number of seasonal influenza cases per sentinel site at the national level in Japan (i.e., seasonal influenza activity) was similar in the preceding 6 epidemiologic years (2014–2019); however, a marked downward trend was observed after 2020, with an estimated decrease of approximately 66%. Notably, employing a standard time-series GLM, with a gamma distribution and logarithmic link function with robust error variances, demonstrated statistically significant immediate and delayed association effects between the overall transmission dynamics of seasonal influenza and the change in adherence to each NPIs; this was true even after accounting for different lag structures, including single and cumulative lag months. Assuming causality, the range of actual changes in the transmission dynamics of seasonal influenza in Japan during the study period (2014–2020) was estimated to range from approximately 4.5% to 1.5% for every average one billion increase in spending on retail sales of hand hygiene products, and from approximately 5.3% to 30.9% for every one million increase in the number of domestic and international airline passenger arrivals, respectively. Furthermore, our main findings did not substantially change and remained robust by sensitivity analyses when extending the period covered to December 2021, suggesting that a long-term temporal association between the transmission dynamics of seasonal influenza and adherence to NPIs had a long-term effect beyond the study period in this study. Assessing such time-series associations using our simple statistical modeling can contribute to successfully (at least partially) capturing the observed decreasing patterns of seasonal influenza activity during the COVID-19 pandemic in Japan, demonstrating that the decrease could be largely explained by the changes in adherence to more NPIs such as hand hygiene measures and travel restrictions. The performance of the proposed model suggests that our framework can provide a plausible proxy for average stochastic variations in the dynamics of seasonal influenza transmission using readily accessible data.

As a remarkable finding, the present preliminary analysis also justified the medium-to-long-series benefits of hand hygiene measures (i.e., approximately 3 months) and described the importance of continuing implementation of adherence to hand hygiene strategies in such a way that supplements existing knowledge. Hand hygiene is a simple, low-cost, and widely adopted NPIs to reduce disease transmission (especially targeting the route of physical contact) and is recommended as a standard precaution for the care of infected persons in the community [44,45]. Indeed, there is mechanistic evidence that hand hygiene inactivates bacteria and viruses [46], and several systematic reviews of observational studies and randomized trials describe that hand hygiene alone is significantly associated with a reduction in many human respiratory diseases in community settings [47,48]. Although much of the literature qualitatively reports the effectiveness of hand hygiene strategies on the dynamics of seasonal influenza transmission, one impressive case–control study reported in Spain successfully quantified the effectiveness of adherence to hand hygiene on the incidence of seasonal influenza infections [49]. Specifically, hand washing more than five times per day was a statistically significant protective factor, suggesting the need for frequent handwashing to prevent seasonal influenza transmission in community settings. Taken together, it is essential to reaffirm the versatile and attractive benefits of adherence to common hand hygiene practices in controlling the spread of seasonal influenza transmission during the COVID-19 pandemic, even considering the uncertainty of effectiveness of hand hygiene [45,47].

Another crucial contribution of the present study is the confirmation of the existence of a decrease in the incidence of seasonal influenza infections at the national level in Japan, following the changes in the number of domestic and international airline passengers attributable to adherence to travel restrictions. Indeed, previous studies have highlighted the effectiveness of travel restrictions owing to the COVID-19 pandemic, providing a valuable opportunity to assess the effect of human mobility behavior changes on other human respiratory infections [50,51,52,53,54,55,56,57]. Previous studies assessing the effectiveness of international travel restrictions by mainland China on the risk of COVID-19 outbreaks in other overseas countries focusing on early 2020 reported that a reduction in export volume could delay the importation of cases to cities not affected by the epidemic [51]. Specifically, it describes an average reduction of 81.3% (95% CI: 80.5–82.1%) in daily exports owing to travel restrictions during the first 3.5 weeks of implementation. Additionally, another study by Nishiura et al. reported that border quarantines (i.e., travel restrictions) may contribute significantly to preventing (or at least delaying) the arrival of the virus, especially in small island countries such as Japan. The religious importance of international human mobility restrictions on the transmission dynamics of infectious diseases has also been discussed [56]. However, only a limited number of studies have reported the impact of domestic travel restrictions on the spread of the virus. A recently published study from Japan quantified the effect of domestic travel restrictions and the spread of COVID-19 in Japan and provided impressive insights into the transmission dynamics of other respiratory infections, including seasonal influenza viruses [57]. Specifically, the infection dynamics showed a relative risk reduction (approximately 35–48%) in most prefectures with travel restrictions, suggesting that the degree of passenger volume has a significant impact on risk reduction and that specific airports and their optimal size may be strongly dependent on the domestic network. It was concluded that there may be a strong dependence on specific airports and their optimal size in the domestic network. Based on our present findings and previous empirical literature, we speculate that strategies to restrict the airline network may reduce the risk of seasonal influenza transmission at certain risk-sensitive airports, resulting in a significant reduction in contact patterns between infected and susceptible individuals.

Although the present study focused on the need to identify the potential effect of adherence to NPIs on the transmission dynamics of seasonal influenza in Japan, several caveats must be noted when interpreting the estimates and applying the present results to the assessments. It should be noted that this study is a preliminary finding describing the association between transmission dynamics of seasonal influenza and adherence to NPIs at the national level in Japan in a long series during 2014–2020, including under the COVID-19 pandemic, based on the analysis of observational data, and is regarded as a type of ecological study in causal inference. That is, our findings are likely vulnerable to confounding, and indeed, the biological mechanisms and natural histories behind both social/behavioral and biological/intrinsic factors (e.g., clinical course of patients in characterizing secondary transmission, increased viral replication, high frequency and dose of pathogen shedding, or some other unknown host–pathogen relationships) have not been assessed. Second, because the present study used retail sales of hand hygiene products and domestic and international airline passengers as alternative indicators of NPIs using published observational data available in Japan, these alternative indicators may not reflect the actual effects of adherence to NPIs. In particular, the observed time-series of monthly retail sales of hand hygiene products do not reflect actual numbers and may be affected by potential variations in economic inflation in Japan over the past decade [58]. Third, several other key driving NPIs that decrease transmission, such as education, voluntary self-isolation, school closures, and voluntary mask wearing, were not considered in the present study because of the lack of publicly available epidemiological data at the national level in Japan [59,60,61,62,63]. In particular, it is expected that the habit of voluntary mask wearing in Japan before the pandemic, which is common East Asia, may have contributed significantly to the reduction in transmission of respiratory infections in communities compared to other countries (e.g., US and European countries); unfortunately, we were unable to evaluate this possibility [64]. Indeed, Cowling and Leung described the accumulation of compelling evidence of masks [65], with one impressive study in Europe describing a 12.0% (95% CI: 7.0–17.0%) reduction in time-varying effective reproduction *R* (*t*), the average number of secondary cases produced per primary case in the presence of interventions and immune individuals in a given time period, associated with the introduction of a policy requiring the wearing of masks in all common and public spaces [66]. Interestingly, a decreasing trend in seasonal influenza activity in Japan was observed around January 2020, before the COVID-19 pandemic (Figure 1). The retail sales of hand hygiene products were low, and domestic and international airline passenger arrivals were high just as the pre-COVID-19 pandemic period. Japanese people may have started the self-protection like voluntary mask wearing from mid-January since the first imported case of COVID-19 was identified on 16 January 2020 in Japan, and the broadcasting media reported sensationally about this newly identified viral pneumonia [53]. Those early behavioral changes may have affected the decreased trend of influenza activity during January 2020. Additionally, other driving factors, such as the adaptive evolution among multiple variants of SARS-CoV-2 as variants of concern (i.e., Alpha [B.1.1.7], Beta [B.1.351], Gamma [P.1), Delta [B.1.617.2], and recently Omicron [B.1.1.529]), viral interference (i.e., one virus being prevented from multiplying by another), founder effects (i.e., variants introduced into populations with locally elevated transmission can increase in proportion at a national level without a bona fide transmission advantage), and immune escape attributable to vaccinations have not been assessed [67,68,69,70,71]. Consideration of these possible driving factors affecting the transmission dynamics of COVID-19 and improving the stability of the estimates is crucial for future studies. Forth, airline passenger arrivals are not the only force driving the spread of the virus transmission [54]. In countries (or prefectures) sharing a land border or separated by a small stretch of water, the effect of sea and land travel on the spread (e.g., via train and road traffic) of the virus could be greater than that of air travel. We propose further detailed research considering transportation data from air, sea, and land travel to accurately capture the observed hypothetical variations in Japan. Fifth, although the current statistical model used average ambient temperature and relative humidity as meteorological conditions, there may be other confounding factors (e.g., rainfall, wind speed, ultraviolet (UV) radiation, and air pollutants including PM_2.5_ and PM_10_) that are associated with the transmission dynamics of seasonal influenza [72,73,74,75]. However, several previous studies have suggested a consistent association between the dynamics of seasonal influenza transmission and average ambient temperature and relative humidity, which may have been a highly explanatory predictor [27,28,29]. Sixth, the epidemiological effects of adherence to NPIs have not been fully quantified. Indeed, we were unable to fully assess the heterogeneity of transmission dynamics in specific regions associated with alternative indicators of each NPI because the present study covers the whole of Japan rather than each prefecture. Indeed, the epidemiological effects of NPIs on the transmission dynamics of seasonal influenza may have different probability distributions, as it is necessary to consider the increase in local clusters of COVID-19 in remote prefectures of Japan [76,77,78,79]. Additionally, compared to severe measures, such as city closures (i.e., lockdown), implemented in other countries, the legal enforcement of the state of emergency declaration in Japan is moderate, and the government relies on people’s voluntary action and the resulting peer pressure to reduce contact at the community level [80]. Differences in the intensity of interventions may have contributed to the beneficial positive effects of hand hygiene strategies and travel restrictions in the present study by characterizing the transmission dynamics specific to Japan. Changes in transmission dynamics due to the intensity of adherence to NPIs may be modified in countries other than Japan, which limits the generalizability of this study. Seventh, because we aimed to quantify the independent effects of NPI indicators on the transmission dynamics of seasonal influenza, we failed to consider effect modification (e.g., interaction terms) using the types and subtypes of seasonal influenza viruses (i.e., A/H1N1pdm09, A/H3N2, B/Victoria-lineage, and B/Yamagata-lineage), sociodemographic factors (e.g., sex, age, and social capital in terms of social epidemiology), and spatiotemporal differences (e.g., Tokyo, Osaka, Hokkaido, and Okinawa with distinct geographical settings). Of note, there is a little possibility that influenza vaccination coverage affected decreased seasonal influenza activity in 2020. According to the Ministry of Health, Labor and Welfare in Japan, the national influenza vaccine production in 2020, a proxy to the vaccine coverage, was approximately 30 million vials, which remained almost the same level as in 2019 in Japan [81]. In addition, the decrease in reported cases of influenza in 2020 may be a false decrease due to the hesitancy to visit hospitals or poor access during the COVID-19 pandemic [82]. However, overall, outpatient visits for all illnesses in Japan fell by only 10% in 2020 compared to that of 2019. Thus, the drastic decrease of influenza activity in 2020 cannot be explained by the decrease of outpatient visits, suggesting a real low level of community circulation [83]. Further detailed studies are needed to independently assess the effects of individual and multiple interventions and their complementarities on host susceptibility. Eighth, we failed to consider temporal uncertainties other than variations in the optimal lag lengths (i.e., optimal delays in effect). Although there are concerns arising from plausible settings of the incubation period prior to the onset of symptoms of notified cases, substantial data describing biological processes and natural history of the virus and host reservoir population dynamics of seasonal influenza transmission have not yet been pursued in depth [20,27,32,33,34,35,36]. Ninth, the unit of the lag structure introduced in this study is by month, which does not allow for detailed time-series variation over shorter time periods (i.e., units of days or weeks). Therefore, the quantification of the estimates of seasonal influenza transmission dynamics according to the shape of the lag structure may be over- or underestimated. Finally, in the present study, we focused on the identification of changes in the incidence of seasonal influenza infections, including those highlighting the role of NPIs; more explicit modeling for the marginal effect of adherence to NPIs of the dynamics and the extension to projection of heterogeneity owing to localized epidemics will be crucial to future studies. Although scientific interest has focused on identifying the transmission dynamics of SARS-CoV-2, the epidemiological assessment of respiratory infections during and after the COVID-19 pandemic is an urgent issue, and the specific application and extent of consideration of more explicit methods to capture epidemic dynamics of seasonal influenza and HRSVs is a key subject for future studies in Japan.

Lastly, further prospective design analyses describing transmissivity changes of the virus attributable to adherence to NPIs are critical to confirm our main findings and to approach explicit causality. Nevertheless, by using only the available national level observational data in Japan, we demonstrated that a time-series ecological modeling design can be exploited to examine the collective (population) impact of adherence to NPIs. We believe that our approach will shed light on the assessment of the combined effectiveness of adherence to NPIs in Japan.

## 5. Conclusions

This study provides empirical evidence of a time-series association between adherence to more NPIs, such as sanitary measures and travel restrictions, and a reduction in seasonal influenza activity during the 2020 COVID-19 pandemic in Japan. Our findings are based on limited but readily available epidemiological data and conventional deductive estimation approaches utilizing explicit ecological time-series modeling, and suggest that the public health interventions for COVID-19 adopted by the Japanese government may have prevented and reduced the dynamics of seasonal influenza transmission. Assessing the future transmission dynamics and risks of seasonal influenza and other respiratory infections, and communicating these probabilities, considering the detailed effects of various public health interventions, will help guide the design of future broad-based infectious disease control measures and provide crucial lessons for other countries worldwide. Indeed, it would be worthwhile to conduct these analyses to identify which components of NPIs were most effective in preventing seasonal influenza and other respiratory viruses and transmission. In this context, this pandemic has created an incredible natural experiment on a global scale, with similar policies enacted globally in diverse social contexts. Additionally, our findings may enable public health agencies to take early actions regarding making better preparations for understanding the stochastic variation in seasonal influenza disease burden attributable to adherence to NPIs to meaningfully control the COVID-19 pandemic and future epidemics in Japan; this will be particularly important when considering vulnerable groups (especially essential host groups), and preparing and developing optimized containment schemes of exit strategies to avoid unexpected future emergencies. Nevertheless, the plausible reasons for the reduction in seasonal influenza infections observed in the present study can still be multifactorial and complementary, related to hand hygiene measures and travel restrictions. Although we were unable to purify the unique effect of each alternative indicator of NPIs alone, in future studies, several measures may be strengthened or relaxed regionally, and the generalization of our findings may be applied by comparing the differences in adherence to NPIs in different countries and regions. Although these strategies for applying common NPIs are effective in reducing localized viral transmission in community settings based on previous empirical studies [7,84,85,86], convincing mechanistic evidence to support their effectiveness has is not yet adequately understood. Therefore, extension to joint multi-country with different epidemiological contexts and long-term studies are needed to clarify causality and complex interactions and to present more robust evidence. Explicit inferential modeling (including statistical and mathematical models) that can realistically describe the behavior of infectious diseases based on intrinsic and extrinsic assumptions to describe the dynamics of pathogen transmission in populations pre- and post-pandemic critically needs to be considered in detail in the future [87]. Finally, it is crucial to develop studies on statistical causal inference in the context of infectious disease epidemiology in Japan.

## Figures and Tables

**Figure 1 viruses-14-01417-f001:**
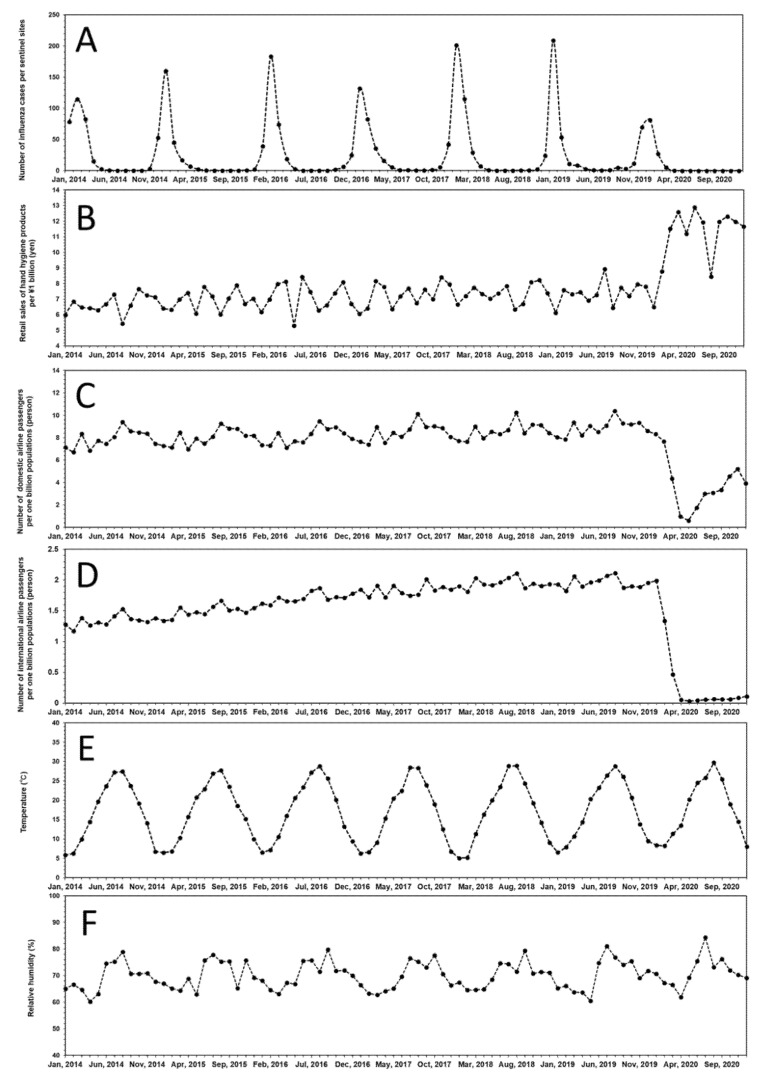
Time-series variations in monthly seasonal influenza activity, alternative indicators of NPIs, and meteorological conditions at the national level in Japan during 2014–2020. (**A**) Monthly seasonal variations in the total number of seasonal influenza cases per sentinel site at the national level in Japan during 2014–2020. (**B**) Monthly seasonal variations in retail sales of hand hygiene products per ¥1 billion (units: yen) at the national level in Japan during 2014–2020. (**C**) Monthly seasonal variations in the number of domestic airline passengers per one million population (units: person) at the national level in Japan during 2014–2020. (**D**) Monthly seasonal variations in the number of international airline passengers per one million population (units: person) at the national level in Japan during 2014–2020. (**E**) Monthly seasonal variations in the average ambient temperature (units: °C) throughout Japan during 2014–2020. (**F**) Monthly seasonal variations in the relative humidity (units: %) throughout Japan during 2014–2020. Detailed numerical statistics by year are described in Appendix A.

**Figure 2 viruses-14-01417-f002:**
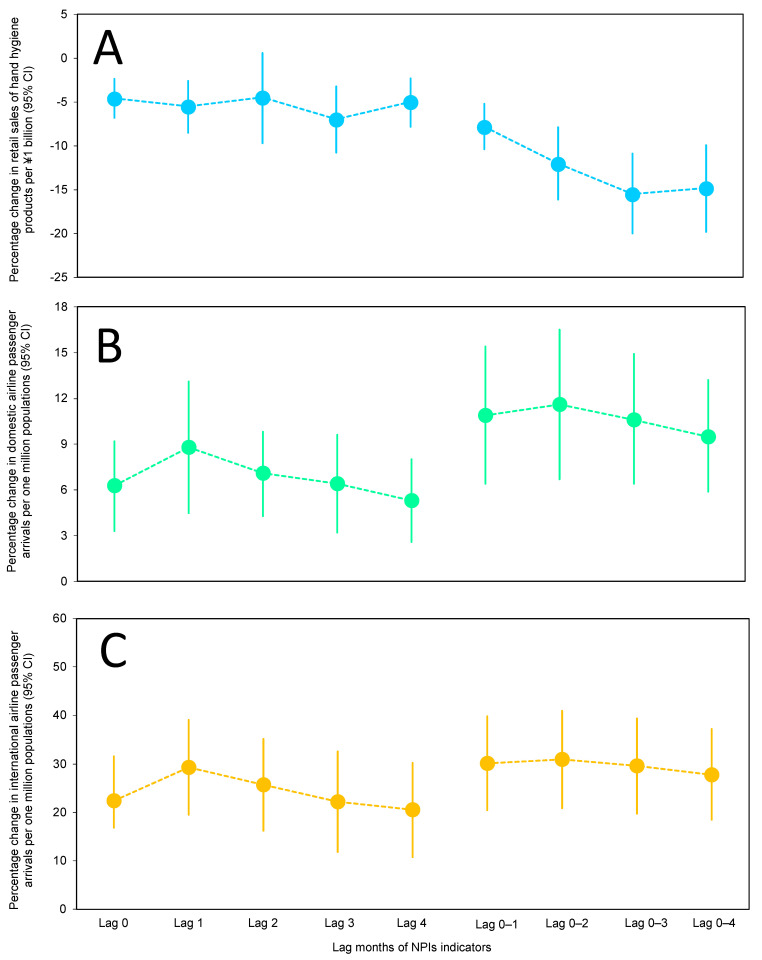
Overall percentage change (mean and 95% CI) in seasonal influenza activity associated with an increase of 1-unit change in alternative indicators of NPIs at different lags at the national level in Japan during 2014–2020. (**A**) Percent change in retail sales of hand hygiene products per ¥1 billion at the national level in Japan during 2014–2020 for monthly seasonal influenza activity. (**B**) Percent change in the number of domestic airline passengers per one million population at the national level in Japan during 2014–2020 for monthly seasonal influenza activity. (**C**) Percent change in the number of international airline passengers per one million population at the national level in Japan during 2014–2020 for monthly seasonal influenza activity. Whiskers show 95% confidence intervals (CIs). The relevant estimates (coefficients, 95% CIs, and *p*-values) for this figure are provided in Appendix A.

## Data Availability

All data used in this analysis are fully documented and freely available from the following GitHub repository: https://github.com/Keita-Wagatsuma/Seasinal-Influenza-activity-during-the-COVID-19-in-Japan (accessed on 25 May 2020).

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
