# Peer review of "Was the Reduction in Seasonal Influenza Transmission during 2020 Attributable to Non-Pharmaceutical Interventions to Contain Coronavirus Disease 2019 (COVID-19) in Japan?"

_viruses, 2022, doi:10.3390/v14071417_

Round 1

Reviewer 1 Report

The author simply analyzed and quantitatively assessed the reduction in seasonal influenza transmission to potential causal effect of various NPIs including hand hygiene products, average ambient temperature, humidity and airline passenger arrivals during COVID-19 pandemic. They find hand hygiene products negatively correlated with seasonal influenza transmission while airline passenger positively correlated with transmission.

Though many factors such as self-isolation, school closures, mask wearing etc. were not analyzed in this study, the authors discussed these factors comprehensively and properly. Meanwhile, the authors discussed the drawbacks of this study comprehensively. In all, this manuscript is well written and may help for seasonal influenza and other human respiratory diseases study.

Reviewer 2 Report

Wagatsuma et al. have analyzed effect of non-pharmaceutical interventions (NPI) on influenza cases. They have considered sales of hand sanitizing agents, number of in-bound passengers, and number of domestic airplane passengers as NPI. Here are some questions authors need to address prior to the acceptance.

  1. The authors used data upto 2020 but it is May 2022. Authors should use the data including 2021.
  2. What is nitrifying in line 101?
  3. Why authors are not taking flu vaccine rates into account?
  4. Authors need to explain why sales rather than quantity of hand sanitizing agents were chosen? How about effect of inflation in the last 10 years?
  5. It is intriguing that number of flu patients did not increase in January 2020 which is before COVID-19 spread in Japan. Neither hand sanitizer nor mobility failed to explain the reduction of flu patients in Jan and Feb 2020.
  6. In Figure 1, X-axis for A-F should be aligned. (plot sizes are different and need to be fixed across A-F)
  7. Why number of cases in 2017 was low?
  8. Isn't number of flu cases during COVID-19 pandemic undercounted due to hesitancy to visit hospitals?

Round 2

Reviewer 2 Report

The authors addressed issues I raised during the previous review round.